# Predictive Model of Preventive Behaviors against COVID-19 in the Older Adult: The PREASOC-COVID-19 Study

**DOI:** 10.3390/ijerph182111067

**Published:** 2021-10-21

**Authors:** Carmen María Sarabia-Cobo, Aroa Delgado Uria, Marta García Lecue, Eva Izaguirre Palazuelos, César Martínez Ruiz, Ángela Fernández-Rodríguez

**Affiliations:** 1Nursing Research Group IDIVAL, Geriatric Nursing Research Group, Faculty of Nursing, Universidad de Cantabria, 39011 Santander, Spain; 2Gerencia de Atención Primaria, Servicio Cántabro de Salud, Cantabria, IDIVAL, Universidad de Cantabria, 39011 Santander, Spain; aroa.delgado@scsalud.es (A.D.U.); mglecue@hotmail.com (M.G.L.); eizaguirre85@gmail.com (E.I.P.); kardes@nodo50.org (C.M.R.); 3CAD Santander, IDIVAL, Gobierno de Cantabria, 39011 Santander, Spain; angela4fero@gmail.com

**Keywords:** COVID-19, sense of coherence, risk factor, coping behaviors, elderly, health knowledge, attitudes, practice, nurses, nursing

## Abstract

Background: The purpose of this study was to analyze the relationship between COVID-19 preventive behaviors, as the dependent variable, with risk perception, coping style and sense of coherence, as independent variables, in older people living in the community. Methods: An observational design for predictive model development. This study was reported following the STROBE statement. The subjects were people over 65 years of age living in the community. Data collection included sociodemographic variables related to COVID-19, risk perception and types, coping styles in the face of contagion, sense of coherence, and preventive behaviors in the face of COVID-19. The data collection period was from November 2020 to January 2021. Results: A total of 305 people participated in this study (71.5% women, mean age 71.34 years; 6.9% suffered from COVID-19 and 44.3% knew someone close to them who suffered from the virus). The coping style variables problem-focused, emotion-focused, and sense of coherence subscales Significance and manageability explained 17% of the variable preventive behaviors against COVID-19. There were statistically significant differences by gender in all subscales, with women scoring higher in all of them; Conclusions: Men with low risk perception, extrinsic risk perception, and low sense of coherence presented worse COVID-19 preventive behaviors. It would be interesting to develop specific prevention and health education campaigns for this population.

## 1. Introduction

The rapid spread of COVID-19 and the severity of the symptoms it can cause in a segment of infected individuals has stretched health systems to the limit [1]. The most vulnerable group and the one that is suffering the highest number of deaths globally, with exorbitant figures, is that of older people [2]. This group is mainly located in the community [3]. Experts worldwide warn that the problem is not going to disappear in the short term and that the security measures adopted by individuals will be key to ensure that the situation remains under control [4,5].

From the epidemiological point of view, it would therefore be necessary to know, especially for the purpose of health promotion and prevention, what the behaviors of older people are towards the adoption of safety measures to avoid contagion, based on their beliefs and attitudes. This is an aspect that the WHO considers to be a priority [6]. Several recent studies in China and Italy have investigated the risk perception and coping strategies followed by older people [7,8]. Their results are conclusive: older people estimate their risk of COVID-19 to be lower than younger people. Women are more concerned about COVID-19 than men.

These data are in line with the literature regarding the health belief model, in which older people, paradoxically, present higher illusory optimism and lower risk perception [9]. These are well-studied facts within processes such as treatment adherence or chronic disease management [10]. The psychological and cognitive processes underlying health belief models, already in place since the 1970s, clearly indicate that constructs such as risk perception, coping styles or adaptation to difficult situations justify the difficulties encountered by health professionals in ensuring good adherence to treatment and adequate control of one’s own health [11]. Of all the health models, this paper will focus on the salutogenic model [12]. This model relates the management of stressful situations (such as this pandemic) to the individual’s capacity for self-management of such situations. It develops concepts such as sense of coherence (SOC) which is directly related to the ability to employ cognitive, affective, and instrumental strategies that help improve the ability to adapt to difficult situations. In health care, the salutogenic paradigm can be developed either for the design of interventions or to reorient health care research [13].

Clearly something has gone wrong with the instructions given to the general population to adopt safety measures in the face of COVID-19 disease progression as for example in Madrid [14]. Recent research points out that health systems have placed more financial effort and resources on hospital and clinical care and have decreased their focus on the community and this has taken a noticeable toll on the containment of the pandemic [15]. Studies related to etiology, clinical control of the disease, the search for valid diagnostic tests and finding an effective vaccine are undoubtedly necessary and important. However, experts already warn that the main key to control is to prevent the onset of the disease and not solely to treat it when its spread can no longer be contained [16,17]. We know that the greatest number of infections occur within the community [1,18]. Therefore, efforts should be made to implement strategies in the community, on behalf of Primary Care services. The Community Nurse is the professional of reference for many older people. If we were to explore older people from the perspective of health belief models, we could design specific strategies to ensure a self-effective control and a realistic coping style in the face of the pandemic [19,20].

It was hypothesized that older people with low risk perception and ineffective coping style are at a higher risk of not adhering to preventive measures [7,8]. However, we incorporated the variable sense of coherence (SOC) and hypothesized that older people with low SOC have lower risk perception, worse coping style, and inadequate preventive behaviors against the risk of COVID-19 infection.

To our knowledge, this is the first study in our country (xx) that specifically measures risk perception and coping styles for COVID-19 disease in community-dwelling older people. Its value is the development of a logistic regression model that analyzes the relationship between COVID-19 preventive behaviors and the variables SOC, risk perception and coping style. This is the first international study, to our knowledge, to carry out a predictive model incorporating the variable sense of coherence together with risk perception and coping style, in community-dwelling older people, within the salutogenic health framework.

The contribution of this study will be an important aid for nurses because, based on the predictive model, we will be able to easily detect those older people with poorer adherence to preventive measures based on their SOC, coping style and risk perception. The results obtained will enable the design of specific health and promotion strategies to favor health behaviors in the face of COVID-19 in older people. The study is led and conducted by nurses, as health agents.

### 1.1. Validity and Reliability/Rigor

This paper has rigorously followed The Strengthening the Reporting of Observational Studies in Epidemiology (STROBE) Statement. Furthermore, the researchers have used structured research instruments. The results are based on larger sample sizes that are representative of the population. In addition, this research study can be replicated or repeated, given its high reliability. Researchers have a clearly defined research question to which objective answers are sought. This project can be used to generalize concepts more widely, predict future results, or investigate causal relationships. The researchers used tools, such as questionnaires or computer software, to collect numerical data.

### 1.2. Aim

The purpose of this study was to analyze the relationship between COVID-19 preventive behaviors, as the dependent variable, with risk perception, coping style and sense of coherence, as independent variables, in older people living in the community.

## 2. Materials and Methods

### 2.1. Design

This study was based on an observational design for predictive model development. The data collection period was from November 2020 to January 2021.

### 2.2. Sample/Participants

The target of the sampling survey was older individuals over 65 years old in XX (north of Spain). We excluded participants who were not cognitively competent, as well as participants who had language impairments, i.e., reading comprehension impairments. A random sample selection was made in Health Centers in XX. The sample size was calculated from the total number of people over 65 years of age in 2020 (128,494) for a confidence level of 95% and considering a beta error of 0.2 for an expected proportion of 80% of potential participants, we estimated a minimum requirement of 127 people. To make a logistic regression model with statistical robustness, in this type of study the literature advises a sample size ten times the number of independent variables to be estimated plus one. The sample size was estimated using the Granmo 7.11 program. Subject selection was randomized following a table of random numbers.

### 2.3. Variables and Instruments

Demographic characteristics were described by categorical variables (sex, age, place of residence and degree of independence using the Barthel Index [21]). 

COVID-19 variables were used based on a dichotomous response (yes/no) to the following questions: Have you suffered from the disease? Has anyone in your close environment suffered from the disease? Has anyone in your close environment died from COVID-19? 

### 2.4. Dependent Variable

* Preventive behaviors scale for COVID-19 (preventive behaviors, PB). This was created ad hoc by a panel of 11 experts (6 nurses specializing in Public Health and 5 physicians specializing in Public Health and Epidemiology). It was developed from the literature [22]. The survey consisted of 19 positive items on preventive behaviors against contagion (wearing a mask, hand washing, etc.), which were answered according to the degree of agreement (1 Strongly disagree to 5 Strongly agree). The factor analysis identified a single factor structure that was called preventive behaviors (Cronbach’s alpha .77). The higher the score, the greater the knowledge and the better the preventive behaviors against COVID-19. The range of scores is from 19 to 95 points.

### 2.5. Independent Variables

* Sense of Coherence was evaluated using the Orientation to Life Questionnaire-13 Items (OLQ-13 or SOC-13) [23] in this work the Spanish validated version [24] was used. The instrument aims to measure a global personality orientation that facilitates adaptive problem solving in stressful situations to which people are subjected throughout life. As in the extended questionnaire, the 13-item questionnaire also measures the dimensions of comprehensibility (with 5 items), manageability (with 4 items) and meaningfulness (with 4 items). The scores obtained express the strength in the sense of coherence of the person, the higher the score obtained, the greater the strength. The answers offer a continuum from agreement to disagreement in 7 response options, represented on a Likert-type scale, from 1 to 7, ranging from "Never", "Rarely" to "Very often" or "Always", both in the positive and negative sense of the question. The OLQ-13 scale has shown good internal consistency, with a Cronbach’s alpha between 0.70 and 0.92 [23,24] and retains the same psychometric qualities as the original 29-item version. Regarding our study, the internal consistency of the items was analyzed using Cronbach’s alpha, which was 0.71, for the comprehensibility subscale it was 0.81, for manageability it was 0.79 and for Significance it was 0.71.

* For the assessment of risk perception, perceived risk factors and coping styles towards COVID-19, a specific questionnaire was designed by a panel of 9 experts (3 nurses specialized in Public Health, 4 psychologists specialized in behavior modification and health education and 2 physicians specialized in Public Health and Epidemiology) based on the literature [25,26,27]. The questionnaire was validated in a sample of 30 elderly, with good psychometric properties and consists of the following three scales:

—Perceived Risk Scale. Consisting of 3 items, in which the person had to show his or her degree of agreement using a Likert scale (0 being No risk and 10 Maximum risk). The maximum score was 30 points, indicating that the higher the score, the greater the perception of risk of infection by COVID-19. The factor analysis identified a one-factor structure which we will call risk (Cronbach’s alpha .735).

—Perceived risk factors for Contagiousness Scale. Composed of 16 items in which the person showed his or her degree of agreement according to a Likert scale (1, Strongly disagree to 5, Strongly agree). The factor analysis identified a two-factor structure (Cronbach’s alpha .781). The two factors identified correspond to how the person perceives the risk factors: as external or environment-dependent risk factors with 9 items (extrinsic risk factor, alpha .721,) or as factors that depend on the individual, with 7 items (intrinsic risk factor, alfa .841). The higher the score, the greater the weight of one risk factor over the other. The range for the extrinsic risk factor is 9–45 and for the intrinsic risk factor it is 7–35 points. The intrinsic factor is desirable because it speaks of the things I can do to protect myself, while the extrinsic factor speaks of the inevitability of the disease and of factors that are beyond my control and over which I can do nothing. The results are consistent with the literature [7,28].

—Coping Styles Scale with Contagion. These were assessed based on 14 items, in which the person showed his or her degree of agreement according to a Likert scale (1 Strongly disagree to 5 Strongly agree). The factor analysis identified a two-factor structure (Cronbach’s alpha .793). The two factors identified correspond to two coping styles in relation to COVID-19: Problem-focused (7 items, alpha .801) and emotion-focused (7 items, alpha .785). The higher the score, the more one risk factor weighs against the other. The range for problem-focused and for emotion-focused is 7–35 points. Of the coping styles that coincide with the literature, problem-focused is preferable, as it is a style based on the active search for solutions [8,29]. 

### 2.6. Data Collection

The nurses participating in the study made a random selection of those persons from their Health Center who met the inclusion criteria. After an explanation of the project, participants signed an informed consent form. Participants completed all variables using an anonymous online survey. To control the questionnaire quality, the same IP address was only allowed to answer once. The questionnaire was designed to be answered from a single IP. In the case of persons without internet access to answer the questionnaire online or who only had a computer/tablet at home to answer the questionnaire, the nurse collected the data via telephone. The data collection period was from November 2020 to January 2021.

### 2.7. Statistical Analysis

The data were analyzed using IBM SPSS Statistics 22 (IBM, Boston, MA, USA), a *p* value of <0.05 (two-tailed) was considered statistically significant. For the descriptive analysis, all variables were analyzed to identify and correct for outliers and missing data. For the analysis of possible missing values, we used the EM (expected maximization) method. We adopted a bilateral contrast and a 95% confidence level. Descriptive statistics (means, standard deviations, and percentages) were used to describe the sample. A comparison was carried out between people who had experienced COVID-19 and those who had not for the variables SOC, coping styles, risk factors and preventive behaviors by means of the Student’s t-test for independent samples. Subsequently, a bivariate correlation analysis was carried out between all the variables in the study using the Pearson’s r test. Only those variables that showed significant correlations with the preventive behaviors scale were introduced in a stepwise, multiple linear regression analysis (MLR) to determine the best predictors. The assumptions of the MLR model were evaluated by means of the following analyses: (1) normality: Kolmogorov–Smirnov test and P–P Normal plots; (2) linearity: partial regression plots; (3) homoscedasticity: scatter plots of typed residuals and typed predictors; (4) independence of errors: Durbin–Watson statistic; and (5) noncollinearity: diagnostics of collinearity (Tolerance >.10 and IVF <.10). Considering the literature on the model of health beliefs and health-generating behaviors, we decided to choose a single predictive model that predicts the high-risk outcome of inappropriate knowledge behavior in the face of contagion risk.

### 2.8. Ethical Considerations

The study was designed according to the ethical standards of the Declaration of Helsinki and received approval from the ethical committee of Servicio xxx de Salud (Approval number: CE INNVAL 20/31) and by the Primary Health Care Management of xx according to the specific national guidelines and conformed to the principles of the Declaration of Helsinki. Prior to data collection, all the participants were provided information concerning the study and signed the informed consent. Participants were informed that all answers would be anonymous.

## 3. Results

### 3.1. Descriptive Analysis of the Sample

A total of 305 people responded (exceeding 35% of the required sample size). Table 1 shows the sociodemographic variables, as well as the questions related to the COVID-19. A total of 67.9% suffered from some chronic disease (hypertension and diabetes mellitus being the most prevalent). The mean Barthel Index was 83.95 ± 8.39.

### 3.2. Preventive Behaviors, Perceived Risk and Risk Factors for Contagiousness Scales

Table 2 shows the mean values for the following scales preventive behaviors, risk and risk factors, by gender and COVID-19 variables. The score obtained on the preventive behaviors scale, which reached a mid-level in the general population (59.26 +/− 4.99 of a maximum score of 95, statistically significant compared to the mean of the scale itself, t = 5.23, *p* = 0.03), was higher and statistically significant in women and in people who had not had the disease, as well as in those who had had someone close to them sick or deceased. The risk scale score was higher than the sample mean (t = 0.12, *p* = 0 .03). Statistically significant differences were found by gender, with women scoring higher. In addition, among people who had had someone sick in their environment, those who had had someone sick scored higher (*p* < 0 .001). The sample analyzed showed a medium perception of the risk of infection by COVID-19 mean (t = 2.36, *p* = 0.02), with a greater weight of the extrinsic risk factor (t = 2.36, *p* = 0.02). There were statistically significant differences in the two subscales of risk factors between those who had experienced COVID-19 and those who had not: those who had experienced COVID-19 scored higher in the extrinsic factor and those who had not had COVID-19 scored higher in the intrinsic factor. There were also differences by gender in the intrinsic factor, with women scoring higher. In addition, also for the extrinsic factor, men scored higher.

### 3.3. Coping Styles Scale with Contagion 

Table 3 shows the results for the coping styles subscales as a function of gender and illness-related questions. The problem-focused coping style scored the highest in the total sample. There were statistically significant differences by gender for both coping styles, with women scoring higher in both, but especially in the problem-focused style. We also found statistically significant differences between those who had suffered from the disease and those who had not for both coping styles, with the emotion-focused style scoring higher in those who had suffered from the disease. Differences were also found between people who had had a death from COVID-19 in their close environment, with those who had had a death from COVID-19 scoring higher in the emotion-focused style. In addition, the same was found for those who had had a patient close to them.

### 3.4. Sense of Coherence in Relation to COVID-19 

When calculating the sense of coherence value for the entire sample, we obtained a total mean SOC of 50.58 ± 11.43 points (out of a total of 91), with the relative order of the dimensions, according to their percentage of each total, being comprehensibility, meaningfulness and manageability (Table 4). Women scored higher in all subscales except comprehensibility, although statistically significant differences were found only in total SOC. Statistically significant differences were found for the SOC Total and comprehensibility subscales between those who had and had not had the disease (higher scores for those who had). Statistically significant differences were also found in the manageability subscale among people who had experienced a death in their close environment.

### 3.5. Predictive Factors of Preventive Behavior against COVID-19 

A statistically significant association was found for age and problem-focused coping style (r = 0.139, *p* < 0.001). However, no association was found between the other sociodemographic variables and the scales studied. Neither was there any association between the COVID-19 variables (having suffered from the virus oneself, someone in the patient’s environment suffering from it and/or someone close dying) and the scales studied. A statistically significant association was found between the variable preventive behaviors and the subscales SOC manageability (r = −0.201, *p* < 0.001), meaningfulness (r = −0.244, *p* < 0.001), and the two subscales of coping styles (problem-focused r = 0.041, *p* < 0.001; emotion-focused r = 0.381, *p* < 0.001. However, not for the remaining scales. For the analysis of the predictive capacity of the preventive behaviors variable of the study variables, Multiple Regression Analysis (MRA) models were applied using the forward stepwise method, after verifying that all the assumptions were met. Given the limitations posed by the sample size for the inclusion of many variables in the regression model, we first performed two MRAs considering the sociodemographic characteristics (age, sex, place of residence and Barthel Index) and the COVID-19 variables (having suffered from the virus oneself, someone close to the patient dying from COVID-19 or having suffered from the virus) as predictors. None of these models were significant: Sociodemographic characteristics (F = 0.22, R2 = 0.001, *p* = 0.1) and COVID-19 variables (F = 0.39, R2 = 0.045, *p* = 0.247). Consequently, the need to include any of these variables as control variables in the multiple regression model of the study variables was ruled out.

A forward stepwise multiple regression analysis was carried out using the preventive behaviors variable as the dependent variable and the SOC manageability and meaningfulness subscales, and the problem-focused and emotion-focused coping styles subscales as predictor variables. The model was significant (F = 35.80; *p* < 0.001) explaining 33.5% of the variance of the criterion variable through the predictor variables problem-focused, emotion-focused, SOC Manageability and meaningfulness. Problem-focused coping style was the most relevant predictor (Beta = 0.20; *p* < 0.001) explaining on its own 17% of the variance of the dependent variable, followed by emotion-focused coping style (Beta= 0.41; *p* < 0.001), SOC meaningfulness (Beta = 0.23; *p* < 0.001) and SOC manageability (Beta = 0.17; *p* = 0.002) (Table 5).

## 4. Discussion

Older adults are the most vulnerable group in terms of morbidity and mortality during the COVID-19 pandemic [30]. Adopting preventive measures against contagion is the most effective measure in addition to vaccination [31]. Exploring risk perception, coping styles and preventive behaviors towards COVID-19 is crucial for the design of preventive strategies within the community. Incorporating another variable such as sense of coherence, within the salutogenic model of health, will help us to understand the underlying mechanisms of adherence to preventive measures. The main findings found in this study were: (i) older people presented average preventive behaviors, especially women and people who had not suffered from the disease and those who had a close relative who was ill and/or died from COVID-19 scored high. (ii) in a predictive model, the two coping styles problem-focused, emotion-focused, and the two subscales SOC Significance and manageability explained up to 17% of the variance of the variable preventive behaviors. These and other relevant results are discussed below.

### 4.1. Overall Results

Older people in our sample presented good knowledge and preventive behaviors towards COVID-19. This is repeated in other similar studies, although the results are disparate [3,32]. There are studies indicating that older people present lower risk perception, due to illusory feelings of invulnerability and illusory optimism, among others [8]. In our study, the values have been average, which does seem to be consistent with their behavior compared to similar previous diseases [33]. In our case it seems that older people have preventive behaviors, although we do not know if this is due to fear [34], or to a greater responsibility compared to other groups of people with a different age or lack of chronic diseases [35]. Women scored higher, which is widely corroborated in other studies [3,7]. It is also notable that people who had not suffered from the disease presented better behaviors than those who had, a finding that speaks highly of the importance of preventive behaviors. The use of different scales largely conditions the ability to compare our findings with other studies. Nevertheless, our scale, preventive behaviors, was based on the literature, was developed by a panel of experts, and showed good properties.

The most predominant risk perception was the extrinsic risk factor. This indicates that the people in the sample perceived that the risk factors depend more on causes external to them (contact with other people, inevitability of the disease, pandemic, etc.) [36]. This seems consistent considering that it is a global pandemic, with numerous infections and deaths. Uncertainty and ignorance of the cause generates a feeling of a lack of control over the situation [37].

In terms of coping styles in the face of contagion, problem-focused scored the highest. It also showed a significant association with age. Older people were more problem-focused. This indicates that older people focused on the problem and how to deal with it to respond to it. The other style, emotion-focused, also with average values, although which were lower, indicated a style more centered on focusing on feelings as a means of avoiding the situation. Women scored statistically higher than men in both styles, for which problem-focused stood out. There is evidence indicating that women present a greater sense of responsibility in the face of health problems, with more preventive and reality-focused behaviors [7,37,38].

The SOC variable presented mean values in all its subscales, with women scoring significantly higher in all subscales except for the comprehensibility subscale. Presenting a medium–high SOC is related to healthier aging, greater adaptation to stress and greater adherence to healthy behaviors [39,40]. This is something which according to the literature is more present in women than in men, although no conclusive explanations for this have been drawn [41]. 

### 4.2. COVID-19 Variables and Scales

Focusing on the differences between people who have had the disease and those who have not, this study reveals interesting findings. Statistically significant differences were found for the subscales of the perceived risk factors. People who had experienced COVID-19 scored higher on the extrinsic factor. This may be related to the fact that those who have been infected may not have strictly complied with safety measures. In part, this may be due to considering that the causes of the virus are alien, extrinsic and therefore uncontrollable [32]. There were also statistically significant differences in coping styles. Those who had suffered from the disease, or who had a family member who had suffered from or died from the virus, scored higher on emotion-focused. It seems consistent to suppose that having been in contact with the disease favors more avoidant coping styles as the person avoids thinking about the problem. These results are similar to other studies [7,37,42]. Regarding the sense of coherence values, we found statistically significant differences between those who had suffered from COVID and those who had not, with higher scores on the comprehensibility and total SOC subscales for those who had been ill. The SOC focuses on the ability to adapt to stressful situations. Comprehensibility refers to the degree to which people make cognitive sense and deal with situations by considering information in an orderly, consistent, structured, and clear manner. It seems clear that having suffered from the disease requires a cognitive structuring of the person to understand what happened, adapt to the consequences, and better understand the situation experienced [39,40].

### 4.3. Regression Model

Undoubtedly, the interesting part of this study is the predictive model. In this model, the two coping styles and two subscales of the SOC, meaningfulness and manageability, stand out as predictors of preventive behaviors. Interestingly, coping styles are the strongest predictors, especially problem-focused. It seems logical that people who focus more on coping and take an active role in preventing the disease also have adequate preventive behaviors [35,43]. Regarding the SOC scale variables, the results seem consistent with other similar findings suggesting that those people who attach high meaning to their life events develop a sense of commitment and involvement with them and this involvement in turn favors them to assume them as structured, predictable, and explainable events [41,44]. In other words, medium–high levels of SOC seem to suggest that they experienced their situations as understandable and meaningful, developing effective preventive behaviors.

### 4.4. Limitations

The results of this study should be interpreted with caution, as the cross-sectional nature of the design prevents us from determining the direction of the relationships between variables. In addition, there are other limitations: all measures were self-report questionnaires; the selection of participants was based on convenience sampling and there was an absence of follow-up data. Although the established theoretical sample size was reached, perhaps a larger participation of subjects would improve the findings of the predictive model.

## 5. Conclusions

To our knowledge, no research has been conducted in Spain analyzing risk perception, coping styles and preventive behaviors against COVID-19 in older people living in the community. Furthermore, no study has examined sense of coherence at a global level, framed within a predictive model.

This exploratory study has enabled us to identify that coping styles based especially on focusing on the problem, as well as the subscales of SOC Significance and manageability, can explain up to 17% of the variance of preventive behaviors against COVID-19 in older people. Identifying elderly, especially men, with low risk perception, in addition to an extrinsic risk perception and low sense of coherence, seems key for implementing specific prevention measures.

The study of risk perception, coping styles, sense of coherence and preventive behaviors in the older population is essential during emergencies to improve guidance regarding the approach and prevention strategies that nurses design in the Primary Care setting.

## Figures and Tables

**Table 1 ijerph-18-11067-t001:** Sociodemographic characteristics of the sample (N = 305).

		N	%
Gender			
	Female	218	71.5%
	Male	87	28.52%
Age		M 71.34	DE 16.2
Place of residence	Rural	83	27.2%
	Urban	222	72.8%
Have you suffered from Covid yourself (confirmed by PCR and/or serology)?			
	No	284	93.1%
	Yes	21	6.9%
Has anyone close to you suffered from the disease?			
	No	170	55.7%
	Yes	135	44.3%
Has anyone close to you died from COVID-19?			
	No	271	88.9%
	Yes	30	9.8%

**Table 2 ijerph-18-11067-t002:** Gender variables and COVID-19 questions by risk scale, perceived risk factors and preventive behaviors (descriptive and differential analysis).

Variable	N	Risk (Range 0–30)	Extrinsic (Range 9–45)	Intrinsic (Range 7–35)	PB (Range 19–95)
		M/SD	M/SD	M/SD	M/SD
Total	305	18.57 * (5.01)	25.32 (4.06)	14.01 (3.88)	59.20 * (4.99)
Gender					
Female	218	18.94 (5.13) *	22.48 (3.95) *	27.48 (2.88) *	68.09 ** (4.77)
Male	87	15.67 (4.62)	29.93 (4.34)	17.37 (2.86)	50.44 (5.23)
Have you suffered from Covid yourself (confirmed by PCR and/or serology)?					
Yes	21	19.18 (5.54)	29.00 * (2.19)	22.14 * (3.38)	40.33 * (4.18)
No	284	18.09 (4.51)	25.50 (4.47)	27.45 (2.87)	68.33 (3.97)
Has anyone close to you ^a^ died from COVID-19?					
Yes	30	17.10 (5.77)	26.63 (3.17)	22.87 (3.42)	59.43 * (4.95)
No	271	18.72 (4.95)	25.15 (4.06)	24.14 (3.08)	48.00 (4.32)
Has anyone close to you a suffered from COVID-19?					
Yes	170	21.58 (4.25) **	25.21 (4.13)	23.05 (2.14)	68.23 * (4.47)
No	135	19.65 (4.26)	25.42 (4.04)	24.08 (3.21)	58.07 (4.41)

* *p* value < 0.05. ** *p* value < 0.001. Extrinsic = extrinsic risk factor. Intrinsic = intrinsic risk factor. PB = preventive behaviors. ^a^. Anyone close to you = family, close friends.

**Table 3 ijerph-18-11067-t003:** Coping styles in the face of COVID-19.

Variable	N	Problem F (Range 7–35)	Emotion F. (Range 7–35)
		M/SD	M/SD
Total	305	27.15 (4.17)	18.27 (3.48)
Gender			
Female	218	30.92 (4.12) **	19.51(5.11) *
Male	87	26.8 (4.25)	17.41 (4.46)
Have you suffered from Covid yourself (confirmed by PCR and/or serology)?			
Yes	21	21.11 (4.87) *	29.60 (3.60) *
No	284	26.44 (3.16)	19.23 (2.46)
Has anyone close to you died from COVID-19?			
Yes	30	28.05 (3.75)	28.58 (2.92) *
No	271	27.76 (3.18)	21.17 (4.83)
Has anyone close to you suffered from COVID-19?			
Yes	170	27.22 (4.23)	27.64 (4.52) **
No	135	24.04 (3.87)	18.86 (3.35)

* *p* value < 0.05. ** *p* value < 0.001. Problem F = problem-focused. Emotion F = emotion-focused.

**Table 4 ijerph-18-11067-t004:** Sense of Coherence (SOC) and subscales.

Variable	N	SOC Total (Range 13–91)	SOC1 (Range 5–35)	SOC2 (Range 4–28)	SOC3 (Range 4–28)
		M/SD	M/SD	M/SD	M/SD
Total	305	50.58 (11.43)	18.58 (5.02)	10.84 (3.51)	12.82 (2.54)
Gender					
Female	218	59.43 (11.37) *	18.48 (4.84)	11.07 (3.40)	12.85 (2.50)
Male	87	48.40 (10.39)	18.84 (5.49)	10.29 (3.74)	12.76 (2.67)
Have you suffered from COVID-19 yourself (confirmed by PCR and/or serology)?					
Yes	21	58.90 (9.40) **	22.86 (2.29) **	12.29 (3.15)	13.48 (1.44)
No	284	49.96 (11.36)	18.27 (5.03)	10.74 (3.52)	12.77 (2.60)
Has anyone close to you died from COVID-19?					
Yes	30	50.63 (9.86)	18.67 (4.27)	9.53 (3.41) *	12.63 (2.39)
No	271	50.77 (11.60)	18.64 (5.12)	11.03 (3.51)	12.86 (2.39)
Has anyone in your close environment suffered from COVID-19?					
Yes	170	51.87 (12.21)	19.30 (4.72)	11.12 (3.55)	12.68 (2.41)
No	135	49.54 (10.69)	18.02 (5.21)	10.63 (3.49)	12.94 (2.65)

* *p* value < 0.05. ** *p* value < 0.001. SOC1 (Comprehensibility); SOC2 (Manageability); SOC3 (Significance).

**Table 5 ijerph-18-11067-t005:** Multiple Regression Analysis Model.

Predictors	Increase in R2	Adjusted Increase in R2	B	Standard Error	Beta	t	Sig.
Problem f.	0.17	0.17	0.12	0.03	0.20	3.61	0.000
Emotion f.	0.06	0.06	0.23	0.03	0.41	7.34	0.000
Significance	0.07	0.06	−0.44	0.10	0.23	−4.16	0.000
Manageability	0.02	0.02	−0.25	0.08	0.17	−3.08	0.002

Dependent variable: Having suffered from COVID-19. Problem f = problem-focused. Emotion f = emotion-focused. Significance = SOC Significance. Manageability = SOC Manageability. R2 total model = 0.337; R2 goodness-of-fit = 0.327 (F = 35.80; *p* < 0.001).

## Data Availability

Not applicable.

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
