# Peer review of "Predictive Model of Preventive Behaviors against COVID-19 in the Older Adult: The PREASOC-COVID-19 Study"

_ijerph, 2021, doi:10.3390/ijerph182111067_

Round 1
Reviewer 1 Report
This is an excellent basic research paper on a topic of great importance to the current discussions on what to do to improve communications regarding the COVID-19 pandemic. The authors have been careful to communicate a very complicated message on the findings of their study. The broad complex problems of how to address behavioral change issues in communication on COVID-19 are clearly revealed and should be noted by those working on these communication issues.
Author Response
Thank you so much!
Reviewer 2 Report
Dear authors:
Congratulations for conducting this research using the health belief model, which is currently of interest due to the COVID-19 pandemic. The investigation has a well-defined research question (Page 3, lines 103-106) which is answered through the methodology and statistical analysis used.
It’s not common to find researchs on the relationship between risk perception, perceived risk factors and coping styles front to COVID-19 pandemic.
Materials and Methods section (Page 3, lines 111-230):
The variables and instruments used are clear and the sense of coherence, the perceived risk scale, perceived risk factors for contagiousness scale and coping styles scale with contagion are well defined, which helps the reader to understand each one of them as well as their score.
2.4 Data collection (Page 5):
For the IP address section (Page 5, lines 199-200) there’s a question:
How did you verify that an IP address responded to a single questionnaire?
i.e., why limited only one response per IP address?, when perhaps in one IP address there could be more than one older adult available to answer questionnaires, since for example in the same IP address could live a married couple of older adults.
Results section table 1 (Page 5-6):
Please include marital status, level of schooling and socioeconomic level of the participants, if they were considered in the questionnaires. This information could be relevant, for avoid biases in the answers.
When referring to "anyone close to you" in the questions asked (Table 1-4):
The term sounds ambiguous and subjective because it’s not known if this person is a family member, a friend, an acquaintance or even a person who lives elsewhere. Please define more clearly "Anyone close to you".
Prevalent diseases in the study group are mentioned but the prevalence of how many older adults were hypertensive, diabetic or if they had any other prevalent disease is not stated (Page 5, lines 236-237).
All the tables provide interesting and medullar information on the results (Good work).
It’s interesting to know that women presented higher scores in all scales (except comprehensibility) (Page 11, lines 379-380).
Finally, the regression model designed identified the two coping styles and two scales of the SOC (meaningfulness and manageability) as predictors of preventive behaviors and specifically the problem-focused coping style (Page 12, lines 407 -410) (Very good work).
Adequate conclusions regarding the analysis of risk and preventive behaviors in the face of the COVID-19 pandemic, which still requires implementation of prevention measures.
Author Response
R: 2.4 Data collection (Page 5):
For the IP address section (Page 5, lines 199-200) there’s a question:
How did you verify that an IP address responded to a single questionnaire?
i.e., why limited only one response per IP address?, when perhaps in one IP address there could be more than one older adult available to answer questionnaires, since for example in the same IP address could live a married couple of older adults.
Authors: This is a good point. The questionnaire was designed to be answered from a single IP. In the case of persons without internet access to answer the questionnaire online or who only had a computer/tablet at home to answer the questionnaire, the nurse collected the data via telephone.
Results section table 1 (Page 5-6):
Please include marital status, level of schooling and socioeconomic level of the participants, if they were considered in the questionnaires. This information could be relevant, for avoid biases in the answers.
A: It is true that these are interesting variables, but previous literature did not indicate their relevance to the specific aspects to be studied, hence they were not collected.
When referring to "anyone close to you" in the questions asked (Table 1-4):
The term sounds ambiguous and subjective because it’s not known if this person is a family member, a friend, an acquaintance or even a person who lives elsewhere. Please define more clearly "Anyone close to you".
A: Clarification in the table.
Prevalent diseases in the study group are mentioned but the prevalence of how many older adults were hypertensive, diabetic or if they had any other prevalent disease is not stated (Page 5, lines 236-237).
All the tables provide interesting and medullar information on the results (Good work).
It’s interesting to know that women presented higher scores in all scales (except comprehensibility) (Page 11, lines 379-380).
A: I do not understand the question. The percentage of response by gender and scale is shown in the tables.
Finally, the regression model designed identified the two coping styles and two scales of the SOC (meaningfulness and manageability) as predictors of preventive behaviors and specifically the problem-focused coping style (Page 12, lines 407 -410) (Very good work).
Adequate conclusions regarding the analysis of risk and preventive behaviors in the face of the COVID-19 pandemic, which still requires implementation of prevention measures.
A:Thank you so much.
Reviewer 3 Report
The research is about a topic of undoubted interest within the current pandemic scenario. In fact, the psychological repercussions deriving from the spread of Covid-19 are multiple and all relevant especially when they are aimed at Prevention, that is, useful in favor of the virtuous population that give rise to self-protection.
The study of the psychic parameters on which a greater or lesser adherence to the Prevention protocols is based is undoubtedly one of the researches worthy of attention as it is of immediate use in preparing the appropriate protocols to be applied in Public Health.
The ways in which the research was conducted appear as the ones studied are specified and distinct and then also be considered in reciprocal interactions to outline the profile of the subject better or weakly adhering to any self-protection procedures.
Basically, it is a research that has the ambition, if extended over time, to identify how it could be possible to intervene on the "mentality" less available to "active" and / or "pro-active" participation aimed at individual defense from 'infection. Individual defense which, as is well known, has enormous repercussions on the general protection of the entire population. As for the tests used, these are well presented and also well thought out. In any case, since it is an experimental study with relevance to the possible benefits obtainable in Public Health, it would be interesting to carry out a replication in the future in other European territories well defined and known in the sociological and epidemiological aspects.
- We suggest to carefully read the text and correct some orthographic errors i.e. : line 22 or line 27 and 61.
- Authors don’t agree with "the instruction given to the general population" at line 64, but they should indicate which country they are referring to, because Bibliografic note is referred to Madrid metropolitan area. It should be indicated since the beginning the geographic area
Author Response
Thank you for your suggestions, we have implemented the changes indicated.
Reviewer 4 Report
Thank you for the opportunity to review the manuscript "Predictive model of preventive behaviors against COVID-19 in the older adult: the PREASOC-COVID-19 study".
In my opinion, this is a novel and interesting study. I don't have any major comment, but only a few minor ones are shown below.
1) The data of the percentage of male shown in Figure 1 is wrong and should be revised.
2) Some yes and no items in table 2, table 3 and table 4 are misplaced in their respective rows, which should be adjusted.
Author Response

(The authors gave the same response as above.)
